# Research on Evaluation Methods for Sustainable Enrollment Plan Configurations in Chinese Universities Based on Bayesian Networks

**Keqin Wang [1,2]**, **Ting Wang [3]**, **Tianyi Wang [3]** and **Zhiqiang Cai [3,*]**

[1] Undergraduate Academic Affairs Office, Northwestern Polytechnical University, Xi'an 710072, China; keqinwang@nwpu.edu.cn
[2] School of Management, Northwestern Polytechnical University, Xi'an 710072, China
[3] Department of Industrial Engineering, Northwestern Polytechnical University, Xi'an 710072, China; wt3377@mail.nwpu.edu.cn (T.W.); wangtianyi@mail.nwpu.edu.cn (T.W.)
[*] Correspondence: caizhiqiang@nwpu.edu.cn

**Abstract:** Evaluation methods based on data-driven techniques and artificial intelligence for the sustainable enrollment plan configurations of Chinese universities have become a research hotspot in the field of higher education teaching reform. Enrollment, education, and employment constitute the three key pillars of talent cultivation in universities. However, due to an unclear understanding of their interconnection, universities have yet to establish robust quantitative relationship models, hindering the formation of an evaluation mechanism for sustainable enrollment plan configurations. This study begins by constructing a relevant indicator system and utilizing real enrollment data from a specific university. Through statistical methods such as correlation analysis, it systematically sorts out key variables and identifies seven effective indicators, including average admission score and first-time graduation rate. Subsequently, by using the increase or decrease in enrollment quotas for each major as the experimental target, evaluation models for sustainable enrollment plan configurations aimed at enhancing the advanced education rate are constructed using naïve Bayes networks and tree-augmented Bayesian networks; these are compared with three other classic machine learning methods. The accuracy of these models is evaluated through confusion matrices and receiver operating characteristic curves. Additionally, the Birnbaum importance analysis method is utilized to prioritize remaining variables, ultimately identifying the optimal combination strategy of indicators conducive to the sustainable development of the advanced education rate. The results indicate that the average admission score, transfer rate, and student/teacher ratio are the top 3 prognostic factors affecting the advanced education rate, with the TAN model achieving an accuracy of 96.49%, thus demonstrating good reliability.

**Keywords:** enrollment plan configurations; sustainable enrollment policies; advanced education rate prediction; importance ranking; indicator combination strategy

## 1. Introduction

Enrollment, education, and employment are vital components of talent cultivation in higher education institutions. To achieve sustainable and high-quality development in higher education, establishing a linkagefeedback mechanism is crucial. Only through the mutual promotion of these three components can a virtuous "closed loop" of development be formed. The enrollment work of a university serves as the starting point of the entire talent cultivation system. High-quality student intake serves as a solid foundation for the quality of talent cultivation and is also a significant guarantee for improving employment rates. The consensus among many educational professionals is to promote enrollment through education and to promote employment through enrollment [1,2].

An analysis of the literature using "enrollment, education, and employment" as the primary keywords reveals the predominant themes of linked mechanisms. Publications have notably surged since 2018, reflecting higher education's shift towards high-quality development. In recent years, many universities have been proactive, urgently seeking feasible solutions, particularly in the field of enrollment planning, with mathematical modeling methods emerging as a prominent focus of attention. Wang explored a linked mechanism for employment-oriented program settings and enrollment plan allocation, effectively enhancing the rationality of university self-construction efforts [3]. Jiang used a combination of subjective and objective weighting methods to assign weights to indicators for professional early warning mechanisms, scientifically and reasonably determining the expansion, cultivation, maintenance, alert, and elimination of various majors, achieving a process of natural selection among majors [4].

From a review of the literature, it can be observed that researchers usually start with qualitative or quantitative analysis to comprehensively evaluate enrollment, education, and employment, and establish linkage mechanisms based on indicator systems. Although it has received widespread attention, the construction of linkage mechanisms in 90% of Chinese universities remains in the exploratory stage. Less than 10% of universities propose leveraging enrollment plans with application rates and employment rates as primary evaluation indicators, constructing red, yellow, and blue alerts for the qualitative analysis of enrollment programs. Fewer use databases to establish mathematical models, quantifying and extracting indicators of enrollment, education, and employment situations and combining them with the actual situation of the school to assign reasonable weights [5]. However, when constructing enrollment plan warning standards, model parameters are often determined through methods, such as surveys, which are relatively subjective and cannot reflect the level of attention appropriate to each indicator.

There are two major challenges: the first challenge includes data collection, integration, and analysis mechanisms, and the second challenge comprises the evaluation of sustainable enrollment plan configurations based on student and education quality. Currently, data related to these challenges are housed in different departments of universities, sourced from various databases, and exhibit significant heterogeneity, making integration challenging. Furthermore, without establishing a comprehensive indicator system, researchers either face the challenge of dealing with high-dimensional data with numerous noisy indicators or struggle to extract key information due to the limited number of selected indicators. Moreover, the task becomes more challenging as most machine learning or deep learning algorithms, while capable of producing satisfactory predictive results, often fall short in establishing dynamic adjustment mechanisms for enrollment plans. For example, despite researchers attempting to utilize Bayesian classification algorithms and incorporating them into decision tree learning models to construct predictions, they have not recognized the unique node relationships of Bayesian networks for studying the states of indicator combinations. Bayesian networks provide a powerful tool for analyzing and optimizing complex relationships. Therefore, it is possible to explore this technology, establish a network structure with the advanced education rate as the target variable, and discover key influencing features under different indicator combinations, subsequently identifying the optimal combination strategy.

In conclusion, with the goal of achieving sustainable improvement in the advanced education rate, it is imperative to construct linkage feedback mechanisms as well as evaluation methods for sustainable enrollment plan configurations.

## 2. Literature Review

### 2.1. Current Research Status of Feedback Mechanisms

Universities are the main actors in formulating enrollment plans, and the linkage between enrollment, education, and employment is indispensable [6]. Anafnova pointed out that the formulation of enrollment plans is a systematic engineering process that requires comprehensive analysis rather than simple quantity adjustments [7]. Wang further

revealed that factors such as socio-economic development environment, school positioning, employment quality, and student quality all influence the formulation of enrollment plans, and these internal and external factors are interrelated [8]. Therefore, to achieve the healthy development of the entire system, it is necessary to integrate and coordinate the three core aspects in universities.

The construction of a linked reform system for enrollment, education, and employment was proposed early on, advocating for high-quality university construction, high-quality graduates, a strong teaching faculty, and teaching quality to promote enrollment. There exists a structural contradiction and a mismatch between the talent cultivation in universities and social demand. University graduates are not necessarily in surplus in terms of quantity but suffer from structural imbalances. Several scholars have analyzed that the root cause lies in the failure to establish a sound linkage mechanism for enrollment, education, and employment. These three aspects are interconnected, and any failure in one aspect will affect the entire education process [9,10].

Additionally, predicting the advanced education rate is an important approach to grasp the effectiveness of enrollment plans and the quality of talent cultivation. Addressing this issue, scholars have conducted in-depth analysis and research from initial expert prediction methods to linear prediction based on time series models and grey system models, and the current mainstream non-linear prediction methods based on machine learning such as neural networks and ensemble learning. However, some issues persist. Despite widespread attention to the interconnection between enrollment, education, and employment, research in this area remains limited in depth. The majority of universities in China are still in the exploration stage, with very few attempting to quantify and extract indicators, combine them with actual school weightings, establish models, and construct early warning systems for enrollment in majors [11–13]. The existing literature shows that the laws governing interconnection remain unclear, with most research focusing on theory rather than empirical studies. The majority of research mainly discusses ideas and frameworks, qualitatively elucidating the relationship between the three components, without establishing quantitative relationship models.

The fundamental task of universities is talent cultivation and achieving a dynamic balance between talent supply and demand. However, there is currently a common issue in universities where the enrollment, education, and employment departments work separately. The importance placed on enrollment as the entry point and employment as the exit point is imbalanced, thus failing to form a virtuous cycle. Through an extensive analysis of the literature, it can be observed that the basic steps of this approach, in sequence, can be summarized as follows [14–16]: establishing an enrollment plan decision support system based on the linkage between enrollment, education, and employment; quantifying and extracting indicators; assigning reasonable weights based on the actual situation of the school; establishing a rational model; calculating the comprehensive quantitative scores of enrollment majors; making qualitative evaluations based on quantification; issuing warnings for majors; and, ultimately, assisting in the construction of a linked mechanism for enrollment, education, and employment in universities.

## 2.2. Current Research Status of Bayesian Networks

Bayesian networks are a type of probabilistic graphical model tool that utilizes graph structures to represent conditional dependencies between variables. Nodes represent random variables, and directed edges represent causal relationships between variables. Probability inference is conducted through the use of joint probability distributions and conditional probability distributions; probability distributions are updated based on known information. Bayesian network technology effectively handles uncertainty and improves model generalization performance by learning dependencies from data. In recent years, significant research progress has been made in fields such as artificial intelligence, data analysis, and bioinformatics, and Bayesian networks have been widely applied in machine learning tasks such as classification and regression.

The application domains of Bayesian networks extend across various industries, encompassing finance, healthcare, and engineering [17–19]. Within the healthcare sector, Bayesian networks find utility in disease prediction, patient risk assessment, and other critical tasks. By amalgamating diverse medical data sources, they can model intricate disease interrelationships, thereby furnishing personalized predictions and diagnostic recommendations for patients. In finance, they are instrumental in risk management and investment decision-making. Through the modeling of market and economic indicators, they facilitate prognostications of future market trends and investment portfolios, aiding investors in devising more efficacious strategies. Within the industrial sphere, they are harnessed for predictive maintenance and production process optimization. By modeling equipment sensor data and production parameters, they enable the early identification of equipment failure risks, consequently enhancing production efficiency.

However, this method is rarely applied in the field of education, especially in the evaluation of sustainable enrollment plan configurations. Bayesian networks provide a powerful tool for analyzing complex relationship networks. For example, by first identifying key variables, a Bayesian network structure for enrollment plans can be established; then, each node can be analyzed in detail to explore the relationship between nodes. Subsequently, statistical studies can be conducted on different combinations to understand the impact of various indicator combinations on the advanced education rate and uncover key features of influential indicators. The advantages of this approach are evident: by probabilistic inference, the uncertainty of enrollment plan configurations can be quantified; Bayesian networks can effectively model the complex relationships involved in enrollment plans, helping to understand the causality between various variables; by uncovering key features of influential indicators, targeted recommendations can be provided to help decision-makers adjust enrollment plans to improve the advanced education rate; models based on Bayesian networks can provide scientific support for educational decision-making, making decisions more data-driven and reliable.

Although Bayesian networks have achieved important results in probabilistic modeling, uncertainty modeling, multi-source information integration, and other aspects in various fields, they still face some challenges such as the computational complexity of complex models, efficiency of parameter learning and inference, scalability of structure learning, integration of domain knowledge, etc. [20]. To improve model performance and efficiency, researchers have been exploring new algorithms and methods, optimizing algorithms for parameter learning and structure learning, and developing approximate inference methods suitable for large-scale datasets.

## 3. Indicator Identification and Corresponding Datasets

In current practices, enrollment plans are often developed independently, to a certain extent, by the admissions department. Hence, the lack of interoperability among multiple information systems within university organizations makes it challenging to coordinate effectively. The ultimate goal of this study is to explore the extent to which factors in the enrollment, education, and employment stages influence the enrollment rate, thereby improving the efficiency of sustainable enrollment plan configurations. To address the information silos within universities, it is imperative to establish a comprehensive set of indicators. Statistical analysis methods will be employed to identify correlations among these indicators, thus establishing a set of effective and reasonable metrics. Subsequently, integrating and storing data from various departments becomes necessary. Constructing a sample dataset for enrollment is also imperative for incorporation into subsequent modeling processes, thereby enhancing the reliability of the models and achieving coordinated development in terms of scale and efficiency.

### 3.1. Construction of an Indicator System

Autonomously formulating enrollment plans is beneficial for the operation of each university, but it also poses a challenge. A practically significant indicator consists of an

indicator name and a numerical value. Regarding the principles for constructing these indicators, Yang proposed five aspects: correlation, scientificity, directionality, guidance, and information flow [21]. Alghamdi, on the other hand, divided the principles of indicators into systematicness, representativeness, comparability, and operability [22].

Enrollment is the initial stage of talent cultivation in universities, with enrollment quality being a fundamental factor influencing plan formulation. Due to the unique nature of the Chinese college entrance examination system, the average admission score and transfer rate are of primary concern. Cultivation constitutes a critical aspect of talent development within universities. The provision of adequate teaching conditions is essential for fostering talent and conducting scientific research. Additionally, the state of discipline construction and the fulfillment of training plans profoundly impact talent development. Employment represents the final phase of talent cultivation at universities and serves as a crucial testing ground. Notably, the employment rate and advanced education are equally significant components. Through comprehensive consideration and analysis, this study examines the quality of enrollment, education, and employment, resulting in the establishment of a system comprising 3 comprehensive indicators and 14 sub-indicators, as presented in Table 1.

**Table 1.** Enrollment, education, and employment indicator system.

| Indicators | Sub-Indicators | Code | Description |
|---|---|---|---|
| Enrollment Quality | Average Admission Score | $Z_1$ | Average admission scores for major. |
| | Application Popularity | $Z_2$ | Application popularity reflects the level of interest and demand. |
| | First Application Rate | $Z_3$ | Individuals apply for a specific major as a primary choice |
| | Transfer Rate | $Z_4$ | The ratio transferring to other majors. |
| Education Quality | Student/Teacher Ratio | $Z_5$ | The student/faculty ratio. |
| | National First-Class Majors | $Z_6$ | Whether it is a national first-class major. |
| | First-Time Graduation Rate | $Z_7$ | The ratio of graduating on the first attempt to the total enrollments. |
| | Course Pass Rate | $Z_8$ | The pass rate in academic performance. |
| | CET-4 Pass Rate | $Z_9$ | The pass rate in the College English Test Band 4 exam. (CET-6 the same) |
| | Academic Warning Rate | $Z_{10}$ | The percentage of undergraduates who receive academic warnings. |
| | CET-6 Pass Rate | $Z_{11}$ | The pass rate in the CET 6 exam. |
| | Competition Winning Rate | $Z_{12}$ | The percentage of individuals receiving awards in competitions. |
| Employment Quality | Advanced Education Rate | $Z_{13}$ | The rate of students pursuing further education domestically and abroad. |
| | Employment Rate | $Z_{14}$ | The ratio of employed and further enrolled students to total graduates. |

The 14 indicators summarized in this paper are comprehensive and detailed, providing valuable insights for subsequent modeling. However, there may be information overlap and irrelevant associations among the indicators, which could pose challenges to the accuracy and reliability of the model. Therefore, it is necessary to conduct data cleaning and preprocessing, followed by statistical analysis and testing of the sample data, to eliminate irrelevant and disruptive indicators and establish a dataset.

### 3.2. Construction of the Dataset

After synthesizing the raw data of the 14 indicators provided by the enrollment, education, and employment departments, we obtained sample values for 44 majors at our university over the past decade. Before establishing the model, it is necessary to analyze and clean the dataset to ensure the quality of the modeling dataset.

Cubic spline interpolation fits cubic polynomials between adjacent data points, ensuring smoothness as the function passes through all points with continuous first and second derivatives [23]. That is why we chose this method to fill in missing values in our dataset. Next, considering that Z-scores are calculated based on the population mean and standard deviation, they are more applicable for general outlier handling situations, especially when dealing with non-large sample sizes. We used this method to detect outliers, transforming any outliers using a logarithmic transformation to minimize their impact. Finally, we normalized the data using MinMaxScaler after the transformation.

A comparison of the sample data box plots before and after data preprocessing, as depicted in Figure 1, illustrates that after data cleaning, the data distribution becomes more compact and reasonable. This enhancement demonstrates the effectiveness and comparability of our data preprocessing approach, which is essential for constructing the Bayesian network model.

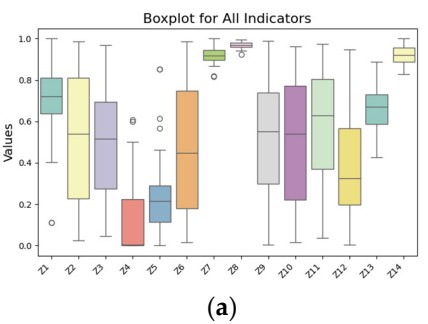

(**a**)

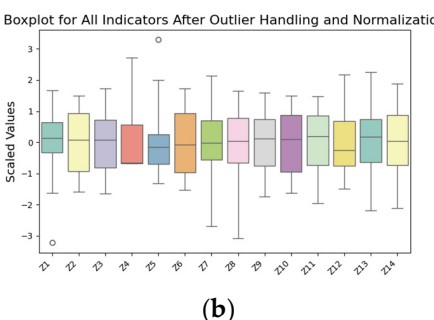

(**b**)

**Figure 1.** Comparison box plot before and after data preprocessing.

To determine the relationships between different indicators, we can conduct correlation analysis. Using a Pearson or Spearman correlation coefficient, we can evaluate the linear or non-linear correlations between various indicators, which helps identify potential patterns or trends. The correlation heatmap is shown in Figure 2.

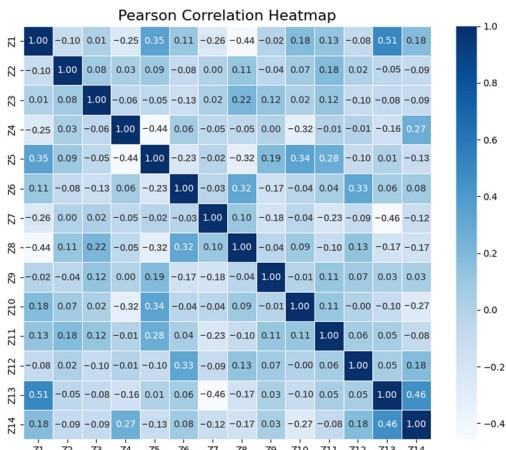

**Figure 2.** Heatmap of correlations among 14 indicators.

By filtering through a significance level (*p*-value) less than 0.05, we found significant correlations among $Z_1$ and $Z_5$, $Z_8$, $Z_{13}$, with a negative correlation with $Z_8$; we also found significant correlations among the following: $Z_4$ and $Z_5$, $Z_{10}$; $Z_5$ and $Z_8$, $Z_{10}$; $Z_6$ and $Z_8$, $Z_{12}$, $Z_7$, $Z_{13}$; $Z_{13}$ and $Z_{14}$; etc. Taking into account the correlations among the indicators, as well as their correlations with $Z_{13}$, we ultimately selected indicators with potential experimental value to include in the modeling system. These selected indicators are as follows: $Z_1$, $Z_4$, $Z_5$, $Z_7$, $Z_8$, and $Z_{14}$. This refined set of indicators will help establish a more effective and targeted model. In summary, the six indicators selected through statistical analysis are used as attribute variables, with the advanced education rate as the object variable and the enrollment quota as the decision variable. The variables for further research are shown in Table 2.

The decision variables typically represent factors that can be adjusted and controlled within the model, influencing its output and playing a significant role in decision-making and prediction. Here, decision variable *D*, representing enrollment quotas, determines if adjustments are needed for specific majors. By controlling enrollment numbers it impacts the study's object variable, the advanced education rate, aiding in evaluating the effects of various enrollment plans.

**Table 2.** Variables included in the future model.

| Variable | Indicators | Code |
|----------|-----------|------|
| Attribute Variable | Average Admission Score | $X_1$ |
| | Transfer Rate | $X_2$ |
| | Student/Teacher Ratio | $X_3$ |
| | First Time Graduation Rate | $X_4$ |
| | Course Pass Rate | $X_5$ |
| | Employment Rate | $X_6$ |
| Object Variable | Advanced Education Rate | $O$ |
| Decision Variable | Enrollment Quota | $D$ |

We can categorize the decision variables of enrollment plan quotas into multiple classes. The increase or decrease in enrollment quotas constitutes discrete variables, with each major varying from a reduction of 17 in its quota to an increase of 30. To ensure appropriate categorization—neither excessive nor insufficient—enabling each category to contain a sufficient amount of sample data and avoiding the problem of imbalanced data while ensuring that the decision variables better support sustainable enrollment plan configurations, we divided them into six categories, using a quota of 10 as the boundary. For example, majors with a reduction of up to 10 in their quotas compared to the previous year were categorized as −1. The classification criteria and categories are presented in Table 3, which effectively reflect different enrollment scenarios.

**Table 3.** Enrollment quota variation classification table.

| Classification Status | Degree Interval | Changes in Enrollment Quota Compared to the Previous Year | Frequency | Percentage |
|----------------------|-----------------|----------------------------------------------------------|-----------|------------|
| −2 | (−10, −20] | Decrease by 11 to 20 | 12 | 0.02% |
| −1 | (0, −10] | Decrease by 1 to 10 | 147 | 0.27% |
| 0 | 0 | No Increase, No Decrease | 122 | 0.23% |
| 1 | (0, 10] | Increase by 1 to 10 | 184 | 0.34% |
| 2 | (10, 20] | Increase by 11 to 20 | 49 | 0.09% |
| 3 | (20, 30] | Increase by 21 to 30 | 24 | 0.05% |

Currently, there is a problem of isolated information islands in domestic universities, where internal information systems and application systems are not interconnected. This situation leads to difficulties in accessing unified enrollment, education, and employment data, as these datasets are non-public and specific to each university. Consequently, there is no unified enrollment data platform accessible to the public. Therefore, this study collected data from the enrollment, education, and employment departments of a specific university, established a unified structured database, and built a data service system that integrates hierarchical cooperation, horizontal integration, and logical unity. The detailed dataset statistics from the database are presented in Table 4.

**Table 4.** Detailed dataset statistics.

| Statistic Index | $X_1$ | $X_2$ | $X_3$ | $X_4$ | $X_5$ | $X_6$ | $O$ |
|-----------------|-------|-------|-------|-------|-------|-------|-----|
| count | 538 | 538 | 538 | 538 | 538 | 538 | 538 |
| mean | 619.2476 | 0.12285 | 7.428285 | 0.969034 | 0.920058 | 0.919034 | 0.653845 |
| std | 0.12224 | 0.095298 | 0.17136 | 0.03571 | 0.02057 | 0.04562 | 0.089363 |
| min | 568.8829 | 0 | 1.269231 | 0.919995 | 0.829694 | 0.817204 | 0.426087 |
| 25% | 596.8511 | 0 | 4.156353 | 0.959338 | 0.888967 | 0.897759 | 0.58722 |
| 50% | 615.5006 | 0.014651 | 6.643368 | 0.969649 | 0.921862 | 0.91852 | 0.671508 |
| 75% | 624.4629 | 0.224199 | 8.50762 | 0.980844 | 0.957335 | 0.945785 | 0.730256 |
| max | 636.69 | 0.609375 | 26.2439 | 0.99384 | 1 | 1 | 0.888889 |

For all variables in the dataset, their values are continuous. However, the later used Bayesian network can only handle discrete variables. Therefore, we applied the K-means

algorithm to partition the values of the variables into two or three ranges. For example, the binary threshold for the object variable was set at 0.638 based on the calculation results. The dataset after discretization is shown in Table 5.

**Table 5.** Discretization of variables in the dataset.

| Variable | Interval | State | Frequency | Percentage |
|---|---|---|---|---|
| $X_1$ | $\leq 609.246$ | 1 | 49 | 9.11% |
| | $609.246 < X \leq 621.251$ | 2 | 232 | 43.12% |
| | $>621.251$ | 3 | 257 | 47.77% |
| $X_2$ | $\leq 0.087$ | 1 | 342 | 63.57% |
| | $0.087 < X \leq 0.296$ | 2 | 74 | 13.75% |
| | $>0.296$ | 3 | 122 | 22.68% |
| $X_3$ | $\leq 6.026$ | 1 | 232 | 43.12% |
| | $6.026 < X \leq 14.27$ | 2 | 257 | 47.77% |
| | $>14.27$ | 3 | 49 | 9.11% |
| $X_4$ | $\leq 0.961$ | 1 | 159 | 29.55% |
| | $0.961 < X \leq 0.977$ | 2 | 208 | 38.66% |
| | $>0.977$ | 3 | 171 | 31.78% |
| $X_5$ | $\leq 0.898$ | 1 | 183 | 34.01% |
| | $0.898 < X \leq 0.947$ | 2 | 208 | 38.66% |
| | $>0.947$ | 3 | 147 | 27.32% |
| $X_6$ | $\leq 0.902$ | 1 | 147 | 27.32% |
| | $0.902 < X \leq 0.943$ | 2 | 171 | 31.78% |
| | $>0.943$ | 3 | 220 | 40.89% |
| O | $\leq 0.638$ | 1 | 220 | 40.89% |
| | $>0.638$ | 2 | 318 | 59.11% |

## 4. Methods

### 4.1. The Technical Roadmap of This Study

The specific research approach of this paper is illustrated in Figure 3.

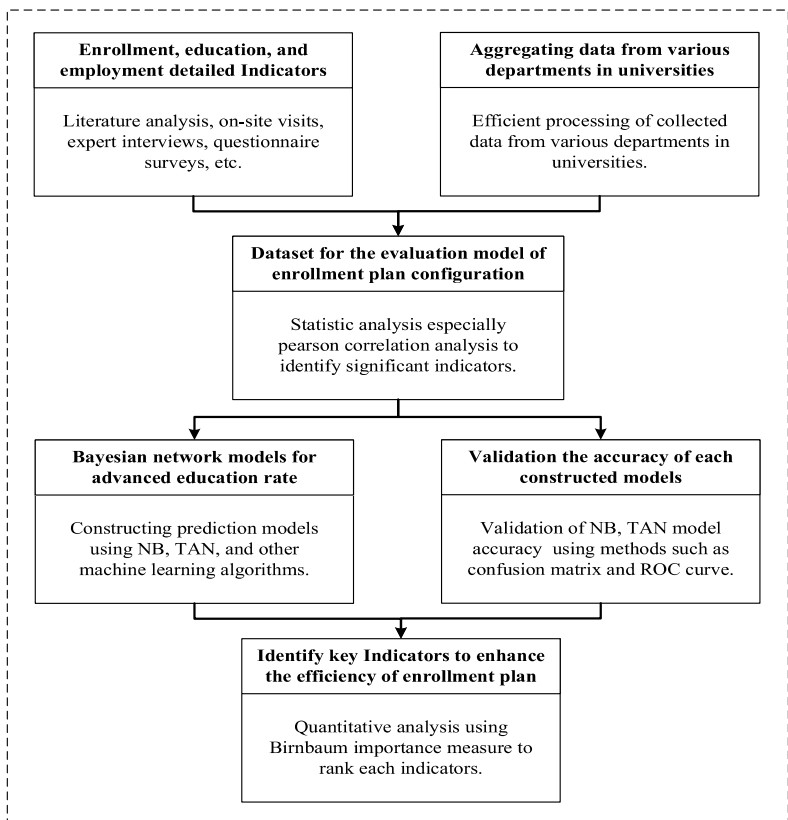

**Figure 3.** Technical roadmap of this study.

### 4.2. Statistical Analysis

Statistical analysis is a process of interpreting, summarizing, and inferring data using mathematical and statistical methods, which includes collecting, organizing, and describing data and then using statistical models and inference methods to gain insights into the underlying patterns of the data. Among these methods, Pearson correlation analysis is a statistical method used to measure the linear relationship between two continuous variables, and its formula is as follows [24]:

$$r = \frac{\sum_{i=1}^{n} \left( X_i - \overline{X} \right) \left( Y_i - \overline{Y} \right)}{\sqrt{\sum_{i=1}^{n} \left( X_i - \overline{X} \right)^2 \left( Y_i - \overline{Y} \right)^2}} \tag{1}$$

where $X_i$ and $Y_i$ are the $i$th observed values of variables X and Y, respectively, $\overline{X}$ and $\overline{Y}$ are their means, and n is the number of observations. The correlation coefficient r gauges the strength of the linear relationship between two variables. Values closer to 1 or $-1$ indicate a stronger linear relationship, while those nearer to 0 suggest a weaker one.

### 4.3. Bayesian Network

The network topology of a Bayesian network is a directed acyclic graph, where nodes represent random variables and arrows indicate relationships. If two nodes are connected, it signifies a causal relationship between them. If one node influences the outcome of another, then a conditional probability value is established. For instance, if node A directly influences the outcome of node B, the directed arc $(A, B)$ is established with an arrow from A to B. The strength of the connection between two nodes is represented by the conditional probability $P(A|B)$. Thus, Bayesian networks classify data samples from a probabilistic perspective, making them an important machine learning method in the field of artificial intelligence. The construction of a network model mainly involves four steps: selecting nodes and variables, determining the network topology, specifying node state probabilities, and determining conditional probability tables between nodes.

The naïve Bayes (NB) classifier is a popular algorithm in Bayesian networks, often used in document classification and spam filtering. It assumes independence among modeling random variables, with each variable related only to one parent node, typically the target prediction variable. The tree-augmented naïve Bayes (TAN) classifier enhances this method by introducing a tree structure to capture dependencies among features, thus relaxing the assumption of attribute independence. Therefore, when dealing with classification problems, especially in cases where correlations exist among features, the TAN classifier method can provide more accurate probability estimates [25,26]. One should suppose that $A_1, A_2 \ldots A_n$ represent $n$ attribute variables, and $C$ represents the target variable. Examples of NB and TAN are illustrated separately in Figure 4.

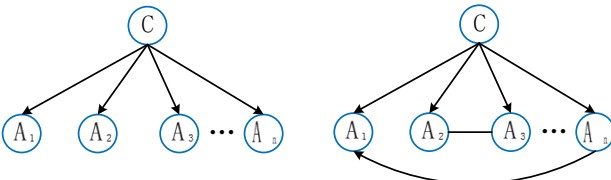

**Figure 4.** NB and TAN classifiers.

The TAN algorithm effectively balances the complexity and learnability of the model, demonstrating, to some extent, the relationships between nodes. This partially compensates for the limitations of the NB network and aids in the analysis of the relationships among enrollment, education, and employment indicators [27]. The algorithm's description process is outlined in Algorithm 1.

| **Algorithm 1.** The Modeling Process of TAN (TAN Algorithm) |
|---|

1. Input dataset: obtain the training instance set D.
2. Compute conditional mutual information: $I_{P_D}(A_i; A_j | C)$ *where* $i \neq j$.
3. Build complete undirected graph : where $A_1, A_2, \ldots, A_n$ are attributes associated with respective nodes. Annotate the edge connecting $A_i$ and $A_j$ with the weight of the conditional mutual information: $I_{P_D}(A_i, A_j | D)$.
4. Build a complete undirected maximum-weight spanning tree.
5. Select root attribute and directionalize the tree: set the direction outward.
6. Add class node: add a node labeled as C and introduce an arc from C to $A_i$.
7. Build TAN model: the resulting directed tree is the TAN model.

### 4.4. Evaluation Methods

The confusion matrix, also known as the classification matrix, is a fundamental method used to evaluate the credibility of prediction models. The possible results generated by the classifier include true positives (*TP*), false positives (*FP*), false negatives (*FN*), and true negatives (*TN*). *TP* and *TN* describe the number of correctly classified instances, *FP* describes the number of negative samples incorrectly classified as positive, and *FN* describes the number of positive samples incorrectly classified as negative. The diagonal elements of the confusion matrix represent the samples correctly classified by the machine learning classifier. The accuracy of the model (*Acc*) represents the proportion of such samples in the total samples, calculated as follows [28]:

$$Acc = \frac{TP + TN}{TP + TN + FP + FN} \tag{2}$$

Recall (*R*), also known as sensitivity, represents the proportion of correctly predicted positive samples among all actual positive samples, as follows [28]:

$$R = \frac{TP}{TP + FN} \tag{3}$$

Precision (*P*) represents the proportion of actually positive samples among all samples predicted as positive. Its calculation formula is as follows [28]:

$$P = \frac{TP}{TP + FP} \tag{4}$$

Two models with low precision and high recall are difficult to compare, and vice versa. To make them comparable, we use the F1 score (*F1*), which ranges between 0 and 1. This metric helps measure both recall and precision simultaneously and represents their harmonic mean, as follows [28]:

$$F1 = \frac{2 * R * P}{R + P} \tag{5}$$

For a machine learning model, it is typically desired to maximize its true positive rate and minimize its false positive rate. However, in real prediction scenarios, both of them usually increase synchronously with the number of positive samples predicted. Therefore, the area under curve (*AUC*) value, which calculates the area under the receiver operating characteristic curve (ROC), is used to evaluate the performance of the model.

### 4.5. Importance Measures

The importance theory is a branch of reliability mathematics that integrates various cutting-edge and hot research areas such as hazard, sensitivity, risk, and importance. It quantitatively expresses the criticality of each component in a system, clearly indicating the differences in criticality and importance among the components within the system.

There are three commonly used methods, including probability importance, structural importance, and key importance. Probability importance, also known as Birnbaum importance, is based on traditional sensitivity analysis methods. It is obtained from the partial

derivative of system reliability with respect to the reliability of individual components, $P_i(t)$, and its formula is typically represented as the following [29]:

$$I^B(i|t) = \frac{\partial h(t)}{\partial p_i(t)}, i = 1, 2, \ldots, n \tag{6}$$

For commonly studied binary systems, importance can be described as the change in the probability of system functioning caused by the change in the state of components. The specific calculation formula is as follows [29]:

$$I(BM)^S_{C_i} = P(S = 1|C_i = 1) - P(S = 1|C_i = 0) \tag{7}$$

where $S = 1$ represents the system being in a functional state, $C_i = 1$ denotes the component being in a working state, and $C_i = 0$ indicates the component being in a failed state. When machine learning algorithms make predictions, they consider the comprehensive states of various indicators, necessitating prioritization among them [30]. This helps identify key predictive indicators, enhancing the credibility and performance of predictive models based on the conclusions and data results.

## 5. Experimental Results

The undergraduate enrollment rate, which refers to the data of students pursuing further education either domestically or abroad after completing their undergraduate studies, is not only consistent with the top-tier innovative talent training positioning of various universities but also serves as an important indicator for evaluating the outcomes of higher education. Therefore, in the modeling process using the aforementioned 538 cases of discretized data, we selected $X_1$, $X_2$, $X_3$, $X_4$, $X_5$, and $X_6$ as 6 node variables, the advanced education rate (denoted as variable $O$) as the object variable, and the enrollment quota (denoted as variable $D$) as the decision variable.

### 5.1. Naïve Bayes Network Model

Based on the theory of NB networks, we set $O$ as the sole parent node in the model, from which seven unidirectional edges were equally drawn, each pointing towards six attribute variables and one decision variable. We partitioned the sample data into training and test sets randomly to facilitate model training and evaluation. The training set comprised 430 records, while the test set consisted of 108 records, adhering to an 8:2 ratio for the separation of the training and testing sets.

The probabilistic graphical model is depicted in Figure 5. Since it is assumed that the attribute variables are independent of each other, the probability of the predicted outcome depends only on the states of each child node. As the probability distributions of the predictive indicators in the child nodes change, the probability of an increase in the advanced education rate in the target parent node also changes accordingly.

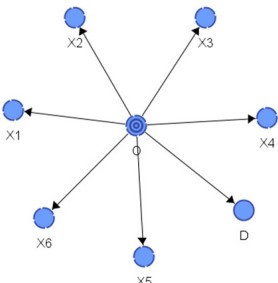

**Figure 5.** Enrollment plan configuration naïve Bayes model.

Specifically, we employed BayesiaLab software (Version 5.0.1) to apply the NB algorithm for modeling purposes. The relationships between each attribute variable and the target variable were derived from prior knowledge and data analysis. Consequently,

we learned the conditional distributions of the edges from the dataset using the software automatically, as illustrated in Figure 6. These distributions pertain to attribute variables, object variables, and decision variables.

| $O$ | 1 | 2 | 3 |
|---|---|---|---|
| 1 | 27.778 | 38.889 | 33.333 |
| 2 | 0.000 | 42.308 | 57.692 |

$X_1$

| $O$ | 1 | 2 | 3 |
|---|---|---|---|
| 1 | 72.222 | 0.000 | 27.778 |
| 2 | 57.692 | 23.077 | 19.231 |

$X_2$

| $O$ | 1 | 2 | 3 |
|---|---|---|---|
| 1 | 44.444 | 44.444 | 11.111 |
| 2 | 42.308 | 50.000 | 7.692 |

$X_3$

| $O$ | 1 | 2 | 3 |
|---|---|---|---|
| 1 | 16.667 | 44.444 | 38.889 |
| 2 | 34.615 | 38.462 | 26.923 |

$X_4$

| $O$ | 1 | 2 | 3 |
|---|---|---|---|
| 1 | 50.000 | 38.889 | 11.111 |
| 2 | 23.077 | 38.462 | 38.462 |

$X_5$

| $O$ | 1 | 2 | 3 |
|---|---|---|---|
| 1 | 16.667 | 44.444 | 38.889 |
| 2 | 42.308 | 38.462 | 19.231 |

$X_6$

| $O$ | −2 | −1 | 0 | 1 | 2 |
|---|---|---|---|---|---|
| 1 | 5.556 | 16.667 | 11.111 | 50.000 | 46.667 |
| 2 | 0.000 | 34.615 | 30.769 | 23.077 | 11.538 |

$D$

| 1 | 2 |
|---|---|
| 40.9 | 59.091 |

$O$

**Figure 6.** NB classifier conditional distributions.

After establishing the NB enrollment plan evaluation model, it is crucial to conduct an initial evaluation of the model's performance based on a test dataset to verify its effectiveness. First, we present the confusion matrix of the model. When the decision threshold of the model was set to 0.638, the overall *Acc* was calculated to be 86.54%, as detailed in Table 6.

**Table 6.** Naïve Bayes model confusion matrix.

| Value | 1 (167) | 2 (371) |
|---|---|---|
| **1 (220)** | 157 | 63 |
| **2 (318)** | 10 | 308 |

Next, we plotted the ROC curve of the model, as shown in Figure 7, and calculated the *AUC* value, as well as the overall *Acc*. Finally, we obtained an area under the ROC curve of 94.63%.

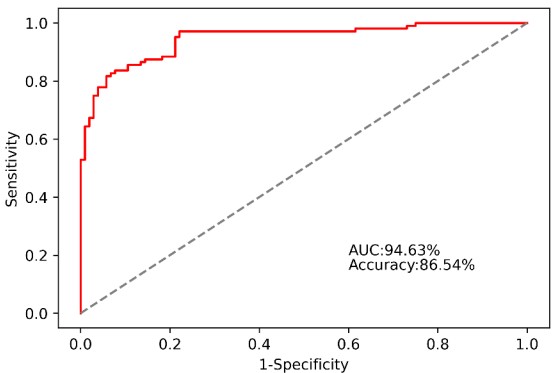

**Figure 7.** Naïve Bayes classifier ROC curve.

### 5.2. Tree-Augmented Bayes Network Model

During the modeling process based on the NB classifier, the correlation between various indicators was not considered. However, in actual enrollment planning, many indicators are often not independent, so the correlation between indicators cannot be ignored. The remaining six indicators were used as node variables for modeling, taking into account the mutual information between each node. Subsequently, a prediction model based on the TAN classifier was generated, as shown in Figure 8.

The TAN network still had only one parent node. However, in addition to the single-directional edges extending to six attribute variables and one decision variable, there were certain correlations between some variables. From the graph, it can be observed that

there is a significant correlation between $X_1$ and $X_2$. This indicates a strong correlation between the two, suggesting that majors with higher average admission scores tend to have lower transfer rates. This is because the arcs generated between nodes using the TAN method cannot only be supported by existing common-sense theories but also uncover some potential dependency relationships between various indicators.

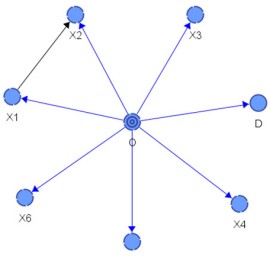

**Figure 8.** Enrollment plan configuration of TAN model.

The conditional distributions of the edges in the TAN algorithm provided are shown in Figure 9, following the same modeling process as that described in Section 5.1.

| $O$ | 1 | 2 | 3 |
|---|---|---|---|
| 1 | 27.778 | 38.889 | 33.333 |
| 2 | 0.000 | 42.308 | 57.692 |
| | $X_1$ | | |

| $O$ | X1 | 1 | 2 | 3 |
|---|---|---|---|---|
| | 1 | 62.500 | 0.000 | 37.500 |
| 1 | 2 | 75.000 | 0.000 | 25.000 |
| | 3 | 100.000 | 0.000 | 0.000 |
| | 1 | 0.000 | 54.545 | 45.455 |
| 2 | 2 | 100.000 | 0.000 | 0.000 |
| | 3 | 100.000 | 0.000 | 0.000 |
| | $X_2$ | | | |

| $O$ | 1 | 2 | 3 |
|---|---|---|---|
| 1 | 44.444 | 44.444 | 11.111 |
| 2 | 42.308 | 50.000 | 7.692 |
| | $X_3$ | | |

| $O$ | 1 | 2 | 3 |
|---|---|---|---|
| 1 | 16.667 | 44.444 | 38.889 |
| 2 | 34.615 | 38.462 | 26.923 |
| | $X_4$ | | |

| $O$ | 1 | 2 | 3 |
|---|---|---|---|
| 1 | 50.000 | 38.889 | 11.111 |
| 2 | 23.077 | 38.462 | 38.462 |
| | $X_5$ | | |

| $O$ | 1 | 2 | 3 |
|---|---|---|---|
| 1 | 16.667 | 44.444 | 38.889 |
| 2 | 42.308 | 38.462 | 19.231 |
| | $X_6$ | | |

| $O$ | −2 | −1 | 0 | 1 | 2 |
|---|---|---|---|---|---|
| 1 | 5.556 | 16.667 | 11.111 | 50.000 | 46.667 |
| 2 | 0.000 | 34.615 | 30.769 | 23.077 | 11.538 |
| | | | $D$ | | |

| 1 | 2 |
|---|---|
| 40.9 | 59.091 |
| $O$ | |

**Figure 9.** TAN classifier conditional distributions.

After establishing the TAN enrollment plan evaluation model, we conducted the evaluation. Firstly, we presented the confusion matrix, still setting the decision threshold to 0.638. At this point, the overall *Acc* was calculated to be 89.42%, as shown in Table 7.

**Table 7.** TAN model confusion matrix.

| Value | 1 (205) | 2 (333) |
|---|---|---|
| **1 (220)** | 184 | 36 |
| **2 (318)** | 21 | 297 |

The plotting of the ROC curve, as shown in Figure 10, ultimately yielded an area under the ROC curve of 96.49%, demonstrating better reliability than the NB method.

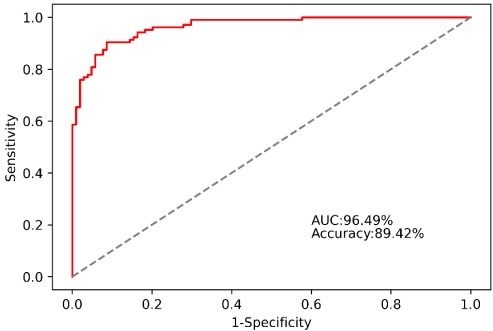

**Figure 10.** TAN classifier ROC curve.

### 5.3. Comparison of Classification Model Performance

To validate the effectiveness of Bayesian networks for evaluating sustainable enrollment plans, we conducted modeling experiments using three machine learning methods: linear discriminant analysis (LDA), logistic regression (LR), and support vector machine (SVM). We then performed comparative experiments and evaluated the models using the *P*, *Acc*, *R*, *F1*, and *AUC*. The data types, training sets, and test sets used for modeling were consistent with the Bayesian network modeling process described earlier. The evaluation results of different classifiers, after experiments based on the test set and five-fold cross-validation, are shown in Table 8.

**Table 8.** Evaluation results of predictions made by different machine learning algorithms.

| Model | *P* | *Acc* | *R* | *F1* | *AUC* |
|-------|-----|-------|-----|------|-------|
| **NB** | 0.8750 | 0.8654 | 0.8585 | 0.8632 | 0.9463 |
| **TAN** | **0.8750** | **0.8942** | **0.9100** | **0.9001** | **0.9649** |
| **LDA** | 0.8713 | 0.8606 | 0.8462 | 0.8514 | 0.9503 |
| **LR** | 0.8738 | 0.8702 | 0.8654 | 0.8672 | 0.9476 |
| **SVM** | 0.8667 | 0.8702 | 0.8750 | 0.8732 | 0.9485 |

For ease of comparison and analysis, we visualized the information from the table, as shown in Figure 11.

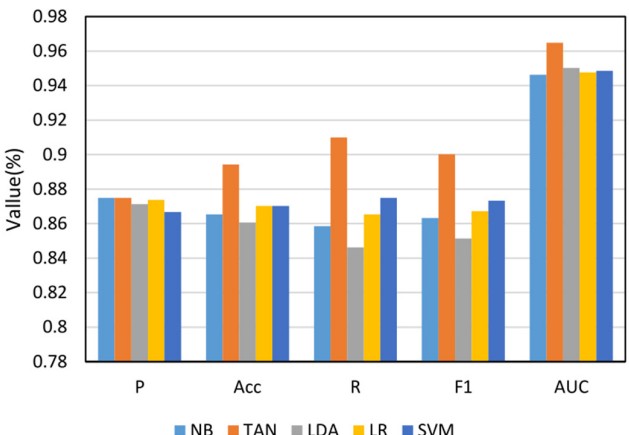

**Figure 11.** Comparative analysis of predictive performance among various methods.

Combining the model evaluation table and the performance comparison chart, we can observe that there was no significant difference between the precision of several models. However, the recall rate of the TAN algorithm is much higher than that of the other three algorithms, reaching 0.91. Since the *F1* is closely related to *P* and *R*, the *F1* is also highest for the TAN algorithm, followed by that of the SVM, LR, and NB algorithms. In terms of *Acc* and *AUC* value, the TAN algorithm still performs the best. Taking into account all indicators, the TAN algorithm performs the best in *P*, *Acc*, *R*, *F1*, and *AUC* value. The NB, LR, and SVM algorithms have relatively mediocre performance, while LDA performs poorly and is not suitable for this type of data research. Therefore, compared with three machine learning algorithms, namely, LDA, LR, and SVM, the TAN classifier demonstrates certain advantages in prediction, as confirmed by the comprehensive consideration of various performance indicators from different perspectives.

### 5.4. Calculation of Influence Factor's Importance

Based on the evaluation model, it is possible to lock the initial state by considering different combinations of states of included indicators and then changing another state to observe the probability distribution of the advanced education rate. To quantitatively analyze the importance of each influencing factor, the Bayesian predictive model can be

combined with importance theory. By altering the type of state for a specific predictive indicator, the difference in prediction rate can be obtained. Additionally, in many cases, a feature's state is not binary but may be divided into low, medium, and high categories. Based on the composite importance measures method for polymorphic systems, the Birnbaum importance of each predictive indicator can be defined as follows [31]:

$$I(BM)_{C_i}^S = \frac{1}{w_i - 1} \sum_{j=1}^{w_i} |P(S = 1|C_i = \text{j}) - P(S = 1)| \tag{8}$$

where $S$ represents the prediction result of $X_1$, and the remaining indicators are denoted by $C$. Each variable has a state w, where $P(S = 1)$ represents the prior probability, and $P(S = 1|C_i = \text{j})$ represents the posterior probability. Based on this method, the importance of each influencing factor is shown in Table 9.

**Table 9.** Birnbaum importance ranking of classification model.

| Prognostic Factors | State | Priori Probability | Posterior Probability | MBM | Rank |
|---|---|---|---|---|---|
| $X_1$ | 0 | 0.1136 | 0 | 1.26785 | |
| | 1 | 0.4091 | 0.6111 | | 1 |
| | 2 | 0.4773 | 0.7143 | | |
| $X_2$ | 0 | 0.6364 | 0.5357 | 0.6627 | |
| | 1 | 0.1364 | 1 | | 2 |
| | 2 | 0.2273 | 1 | | |
| $X_3$ | 0 | 0.4318 | 0.5789 | 0.84895 | |
| | 1 | 0.4773 | 0.619 | | 3 |
| | 2 | 0.0909 | 0.5 | | |
| $X_4$ | 0 | 0.3182 | 0.7857 | 0.879 | |
| | 1 | 0.4091 | 0.5556 | | 6 |
| | 2 | 0.2727 | 0.4167 | | |
| $X_5$ | 0 | 0.2727 | 0.75 | 0.9028 | |
| | 1 | 0.4091 | 0.5556 | | 5 |
| | 2 | 0.3182 | 0.5 | | |
| $X_6$ | 0 | 0.3409 | 0.4 | 0.91075 | |
| | 1 | 0.3864 | 0.5882 | | 4 |
| | 2 | 0.2727 | 0.8333 | | |

Through experimentation, we found that the importance of the six scale features—average admission score, transfer rate, student/teacher ratio, employment rate, course pass rate, and first-time graduation rate—decreases sequentially and that all scales are at relatively high levels. It can be inferred that the first three indicators play important roles in the sustainable improvement of the advanced education rate. The reasons behind this are that higher average admission scores, lower transfer rates, and lower student/teacher ratios in various majors represent a higher quality of student intake and education, leading to more apparent educational outcomes and higher chances of successful advanced education. This experimental result is reasonable; there is a strong positive correlation between average admission scores and advanced education rates, while transfer rates and student/teacher ratios are negatively correlated. Therefore, the strategy we propose is to increase the weights of strongly correlated indicators and increase enrollment quotas for majors with high average admission scores, low transfer rates, and low student/teacher ratios. This experimental result underscores the significance of strategic interventions aimed at enhancing the quality of enrollment and education, thereby fostering improved outcomes in advanced education rates.

## 6. Discussion

Establishing a sustainable enrollment plan evaluation model can comprehensively account for the importance of each stage in enrollment, education, and employment; streamline the linkage effects between each stage; help schools improve current plans; and

complete subsequent plan formulations. Although research on the feedback mechanism has received widespread attention, some issues persist. Previous researchers have attempted Bayesian classification algorithms and integrated them into decision tree models [32], but they did not realize that the unique node relationships of Bayesian networks could demonstrate the states of indicator combinations.

Therefore, we established a Bayesian network structure with the advanced education rate as the object variable and enrollment quotas as the decision variable. By statistically analyzing the states under different indicator combinations, this structure explores the impact of changes in enrollment plans, identifies key influencing features, and identifies the optimal combination strategy conducive to improving the advanced education rate. The Bayesian network model validation confirms that the significance of the six scale features—average admission score, transfer rate, student/teacher ratio, employment rate, course pass rate, and first-time graduation rate—decreases sequentially. The modeling process reflects, to some extent, the real educational environment.

This study primarily focuses on Chinese universities, where the enrollment system, educational structure, and employment market differ from those in other countries. Due to the confidential nature of enrollment data in major Chinese universities, comprehensive data collection for research purposes is challenging. Therefore, we selected the author's affiliated university as the sample. Although the Bayesian network model performed well on this dataset, future efforts should aim to expand data sources. Furthermore, the model itself has limitations such as assuming conditional independence between specific variables, which may not hold true in practice.

However, this does not imply that our model has a regional limitation. Establishing a Bayesian network predictive analysis model allows for a quantitative analysis of factors, facilitating the optimization of current enrollment plans and the formulation of guidance for future plans. Each university can clarify the indicators affecting enrollment plan formulation by considering its national, social, and regional context, as well as its own development status. Then, based on our scheme, key indicators can be selected to establish Bayesian network models, optimize parameters, and propose the optimal combination strategy of indicators aimed at improving the advanced education rate. The broad applicability of the model developed in this study enables its use as a reference for enrollment planning in other universities. Additionally, adjusting parameters based on practical feedback and iteratively optimizing the model further enhances its utility.

### 7. Conclusions

To analyze the linkage mechanisms and explore how to evaluate sustainable enrollment plan configurations, this study utilized real enrollment data from majors at a university over recent years. Firstly, the challenge of collecting, integrating, and analyzing data from multiple departments, coupled with the absence of a comprehensive indicator system, presents difficulties when handling high-dimensional data or extracting key information. This study addressed this challenge by identifying key variables across three stages and conducting correlation analysis. Seven effective indicators—the student/teacher ratio, transfer rate, average admission score, employment rate, first-time graduation rate, course pass rate, and advanced education rate—were selected.

Secondly, faced with the challenge of evaluating the configuration of enrollment plans based on student and education quality, this study selected the Bayesian network model. The advanced education rate was used as the object variable, and the enrollment quotas were used as the decision variable. This study then explored the mapping relationship from enrollment, education, and employment big data to enrollment plan configuration, evaluating model accuracy through confusion matrix and ROC curve analysis. Comparative experiments were conducted with NB, LDA, LR, and SVM machine learning models, with the TAN model achieving the highest accuracy at 96.49%.

Finally, using the Birnbaum importance analysis method to prioritize the remaining variables, this study explored the impact of changes in enrollment plans, under different

indicator combinations, on advanced education rates. The results showed that the average admission score, transfer rate, and student/teacher ratio have extremely significant effects on improving the advanced education rate. In the future, the model can be extended to universities outside of China. Efforts will be made to enhance data collection and analysis mechanisms, refine key indicators for a more accurate evaluation of enrollment plans, further optimize the Bayesian network model, explore more effective parameter optimization methods, and conduct additional empirical research to accumulate practical experience. Strengthening collaborations with international institutions will also be pursued to provide targeted policy recommendations for optimizing enrollment policies, thereby promoting the sustainable development of the higher education system.

**Author Contributions:** Conceptualization, K.W. and T.W. (Ting Wang); methodology, Z.C.; software, T.W. (Tianyi Wang); validation, Z.C. and K.W.; formal analysis, T.W. (Ting Wang); investigation, Z.C.; resources, K.W.; data curation, K.W.; writing—original draft preparation, K.W.; writing—review and editing, T.W. (Ting Wang); visualization, T.W. (Tianyi Wang); supervision, Z.C.; project administration, K.W.; funding acquisition, T.W. (Tianyi Wang) All authors have read and agreed to the published version of the manuscript.

**Funding:** This research was supported by the Key Research Project on Undergraduate and Higher Continuing Education Teaching Reform in Shaanxi Province [Grant No. 21BG007]; the Key Research Project on Educational Teaching Reform at Northwestern Polytechnical University [Grant No. 2022JGWG03].

**Institutional Review Board Statement:** Not applicable.

**Informed Consent Statement:** Not applicable.

**Data Availability Statement:** The experimental data used to support the findings of this study are available from the corresponding author upon request.

**Conflicts of Interest:** The authors declare no conflicts of interest.

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
