# Peer review of "Research on Evaluation Methods for Sustainable Enrollment Plan Configurations in Chinese Universities Based on Bayesian Networks"

_sustainability, doi:10.3390/su16072998_

Round 1

Reviewer 1 Report

Comments and Suggestions for Authors

The manuscript presents an evaluation model for enrollment plan allocation, with a focus on optimizing indicator strategies to enhance higher education rates. The framework and writing are commendable. However, before considering this paper for publication, certain critical aspects require resolution. Addressing these issues thoroughly could significantly elevate the paper's contribution to higher education research. The key concerns are as follows:

Table 1(a) displays outliers in its box plot, whereas Table 1(b) shows these outliers largely eliminated. The rationale behind this discrepancy needs clarification.

A major issue arises in Figure 2, which depicts the correlations among 14 indicators. The narrative in rows 223-235 indicates a selection of indicators based on strong correlations with Z13, specifically including Z1, Z4, Z5, Z7, Z8, and Z14. However, the correlation between Z5 and Z13 is notably weak (0.01), and the correlations for Z4 and Z8 with Z13 are only 0.16 and 0.17, respectively. These figures contradict the stated criterion of strong correlation. The paper would benefit from a more detailed explanation of the selection criteria for these indicators.

For coherence and ease of comparison, it is suggested that the order of the six selected indicators in Table 2 aligns with their presentation in Table 1.

Consistency in terminology is recommended. For instance, the terms "student sources" and "Enrollment Quality" are used interchangeably in the text and Table 1, yet they convey the same concept. Uniformity in language would enhance clarity.

The representation of 'Average Admission Score' as Z1 in Table 1 and X3 in Table 2 is confusing, especially considering the percentage values associated with X3 in Table 4. Clarification is needed on whether this percentage refers to the proportion of students exceeding the admission score threshold.

A grammatical adjustment is needed in line 545, where "rate" should be pluralized to "rates" to fit the context.

Reviewer 2 Report

Comments and Suggestions for Authors

Forming a university admission plan is a complex task that takes into account a number of factors, including the forecast of graduates’ employment. In addition, the admission plan is influenced by a number of external factors, including proximity of other universities with similar specialties. If the university is funded by the state, then the state can regulate a number of parameters of the university, for example, the ratio of students to teachers. Therefore, the authors should explain in more detail the technology of the admissions campaign in China in general terms or provide an example of their university experience, since it is doubtful that the university admissions plan is often developed by the admissions committee independently (line 167). In addition, all data were collected from only one university. In order to provide a comprehensive analysis of article’s outcomes in reference to other universities, it is necessary to characterize it in more detail how typical this university is for the education system in China.

The research objective formulated in the introduction to part 3 is relevant (lines 169 -172): explore the extent to which factors in the enrollment, education, and employment stages influence the enrollment rate. However, as a result, the authors propose the following strategy: increase enrollment quotas for majors with high average admission scores, low transfer rates, and low student-teacher ratios. But, firstly, this strategy does not include employment of graduates at all. Secondly, is it necessary to conduct such serious research for such a trivial result?

At the end of the article, the authors describe creating reliable models for predicting the level of further education and a model for assessing the admission plan. However, the paper only applies well-known machine learning models to conduct experiments. Therefore, the novelty here lies only in the database collected by the authors.

Now let's move on to specific comments regarding the content of paragraphs 3-5:

1. In section 4.1 in the roadmap (Figure 3), the authors state that they used the principal component analysis method to determine significant indicators. However, in the classic version, PCA creates new artificial variables and does not include the assessment of the significance of the original indicators, nor feature selection as it is mentioned in 4.2, line 289. It is necessary to clarify how PCA was used to determine significance. Also in 3.1(lines 236-238), the authors write that they identified 3 main components with the PCA method. Judging by the further description of the models, these components are not used anywhere. It is necessary either to explain how the main components were used, or to remove the text about PCA from 3.1 and from 4.2.

2. Line 317: Do the authors mean the target variable (response variable) when they refer to the "classification variable"?

3. According to the description of the response variable Advanced Education Rate presented in Table 1, it appears to be a real number, i.e. a numerical variable, not a categorical one. It is not clear why the authors are building a binary classifier to predict the values of this variable. Why not to use regression, for example? It is also unclear how the response variable was binarized. It is necessary to specify the rule used for binarization and justify the chosen threshold value for binarization.

4. In paragraph 5, there is not enough information provided about the Bayesian models. It is necessary to specify how the conditional distributions for the edges of the graph were obtained. If they were obtained through modeling, it is necessary to indicate which distribution family was used. In Figure 7, for the TAN model, there is only one edge connecting variables X_i. It is not clear why there are no connections between other nodes. The process of constructing the corresponding graph needs to be described.

5. It is necessary to provide information on how the data was split into the training and testing sets, and how many records were included in each of them.

Reviewer 3 Report

Comments and Suggestions for Authors

The study focused on assessing models for enrollment plan configuration that were created utilizing both Naive Bayes networks and tree-augmented Bayesian networks. The main strengths of the paper are a good literature review and a relevant task. In this paper, actual enrollment data from diverse majors at a university in recent years is employed. The paper identifies crucial variables through three stages and performs correlation analysis. Next, the paper investigates the mapping relationship between enrollment, education, and employment big data to enrollment plan configuration. It systematically examines the application of Bayesian networks in predicting enrollment plans.
Minor comments
1. The authors do not justify enough why the Bayesian Networks were chosen for the research. I propose to pay more attention to this in the introduction.
2. The authors are suggested to add detailed dataset statistics in Section 3.
3. The results presented in Table 9 and in Figure 9 should be more discussed in the paper. Please highlight the best results in Table 9.
4. Please discuss the limitations of the models used.
Other comments
1. Reference [29] -- the initial letter of the name is missing.
2. The citations in the text are printed manually. Please replace them with \cite{}

Reviewer 4 Report

Comments and Suggestions for Authors

The manuscript presents evaluation methods for enrollment plan in Chinese Universities. The reviewer has the following comments/concerns:

1) The presentation of the paper is poor. Sentence construction and language must be improved. The paper is not fully comprehensible in its current format.

2) The abstract of the paper should be revised.

3) The introduction does not provide the proper gap analysis to justify the use of such scheme/approach. The authors have to clearly mention what is missing, rather than making generic statements such as "most of ...."

4) While the paper should target Enrollment Plan, the indicators presented in Table 1 comprise of only 4 indicators for enrollment and all others indicators map to Education and Employment. The authors keep discussing these as well throughout which makes the contribution vague. They should clearly focus on Enrollment indicators and methods

5) The paper focuses on Chinese universities, but at some point, the authors should discuss why they keep their focus on a limited area, and what would be the change in model/study if the scheme is applied elsewhere in the world.

6) The authors have to justify why they used Bayesian Networks

Comments on the Quality of English Language

The paper is full of language errors, especially the abstract which must be revised.

Reviewer 5 Report

Comments and Suggestions for Authors

Dear Authors,

First of all, I would like to thank you for your efforts in preparing and submitting your manuscript. The paper evaluates the enrollment plan configuration in Chinese universities by utilising Bayesian networks. The topic is interesting, however, I have the below concerns for your consideration:

1- What is/ are your research question(s)? How did you conduct your literature review (the procedure you followed and the boolean operators you utilised)? What is/ are the research gap(s)?

2- Please justify the sub-indicators in a scientific way (i.e. is there any other sub-indicator?). In addition, the description of the sub-indicator #14 is vague (i.e. What do you mean by "further education situation"?). Also, please clarify the term "CET", I believe it refers to "common entrance test", but this should be clarified to the readers.

3- In line 215, you mentioned that you did use Z-score, please justify that to the readers in a scientific way (i.e. someone thinks to use t-score).

4- In line 282, you mentioned that you did utilise PCA, that is fine, but how did you avoid the weakness of Q-static plot? Please clarify that.

5- In line 294, your main technique is the Bayesian networks, that is significant, but you need to justify your choice in a scientific way (i.e. someone thinks to use decision tree).

6- Table 2, D needs to clarified, please elaborate.

7- Please expand the discussion part and link it to your research questions and gaps.

8- Please suggest future research agenda within the conclusion part.

Thanks again, I look forward to your modified version.

Best of luck!

Comments on the Quality of English Language

Moderate

Round 2

Reviewer 1 Report

Comments and Suggestions for Authors

OK, the concerns have been addressed.

Reviewer 2 Report

Comments and Suggestions for Authors

We can say that all the comments, except for remark 3, were taken into account by the authors, although their justification for conditional distributions in the Bayesian network (response to remark 4) does not look very convincing (the authors write that they know this from past studies, but they do not provide links)

But the response to remark 3 remained unanswered. I would especially like to understand why exactly 0.638 was taken as the binarization threshold. Maybe this is some kind of criterion for the national education system: if the percentage of those continuing their education is more than 63.8, then this can be considered a success of the educational program? Most likely, this value was chosen to achieve the best metric values. However, I believe that the article can be published in this form.

Reviewer 3 Report

Comments and Suggestions for Authors

I am pleased to report that the authors have addressed all of my comments. The presentation of the material in the article has improved significantly. I believe the article is now can be accepted.

Reviewer 4 Report

Comments and Suggestions for Authors

The concerns are addressed. I have no further comments

Comments on the Quality of English Language

None specifically

Reviewer 5 Report

Comments and Suggestions for Authors

Dear Authors,

Thanks for your efforts in revising your manuscript. The paper can be processed, but please address the below point:

* Each formula should be supported by a reference (its source).

All the best!
